# Intratumoral Heterogeneity of Expression of 16 miRNA in Luminal Cancer of the Mammary Gland

**DOI:** 10.3390/ncrna6020016

**Published:** 2020-05-11

**Authors:** Yuliya A. Veryaskina, Sergei E. Titov, Vlada V. Kometova, Valerii V. Rodionov, Igor F. Zhimulev

**Affiliations:** 1The Federal Research Center Institute of Cytology and Genetics, Siberian Branch of the Russian Academy of Sciences, 630090 Novosibirsk, Russia; 2Institute of Molecular and Cellular Biology, Siberian Branch of the Russian Academy of Sciences, 630090 Novosibirsk, Russia; titovse78@gmail.com (S.E.T.); zhimulev@mcb.nsc.ru (I.F.Z.); 3AO Vector-Best, 630117 Novosibirsk, Russia; 4National Medical Research Center for Obstetrics, Gynecology and Perinatology named after Academician V. I. Kulakov of the Ministry of Healthcare of the Russian Federation, 117997 Moscow, Russia; vladastasiatema@mail.ru (V.V.K.); V_rodionov@oparina4.ru (V.V.R.)

**Keywords:** biomarker, breast cancer, miRNA, luminal breast cancer, intra-tumor heterogeneity

## Abstract

The purpose of this work is to determine the intratumoral distribution of miRNA expression profiles in luminal breast cancer (BC). The study included 33 certain BC cases of the luminal A or luminal B (Her2-) subtypes. The relative expression levels of miRNA-20a; -21; -125b; -126; -200b; -181a; -205; -221; -222; -451a; -99a; -145; -200a; -214; -30a; -191; and small nuclear RNAs U6, U54, and U58 were measured by RT-qPCR in four intratumor areas in each of 33 luminal BC specimens and in surrounding normal mammary gland tissues. Comparative analysis of miRNA expression levels between normal mammary gland tissue and different intratumor areas revealed that only four miRNAs (miRNA-21, -200b, -200a, -191) appear as consistently differentiating markers. A comparative analysis of miRNA expression levels between normal mammary gland tissue and the tumor border revealed statistically significant differences for ten miRNAs; 10 miRNAs show differential expression between normal mammary gland tissue and central tumor specimens; 9 miRNAs show differential expression between normal mammary gland tissue and tumor periphery 1; 13 miRNAs show differential expression between normal mammary gland tissue and tumor periphery 2. After comparing the tumor periphery 1 and tumor center, we found statistically significant differences in expression between five miRNAs and after comparing the tumor periphery 2 and tumor center, differences were observed for 12 miRNAs. MiRNA expression levels are subject to considerable variation, depending on the intratumor area. This may explain the inconsistency in miRNA expression estimates in BC coming from different laboratories.

## 1. Introduction

Breast cancer (BC) is the most common type of cancer among women worldwide. Breast carcinomas are divided into two classes: monogenic and polygenic. Each monogenic tumor appears to contain one large clonal subpopulation with a highly stable chromosome structure. Polygenic cancers contain several clonal tumor subpopulations each [1]. Yates et al. published their results on the spatial distribution of subclones for 12 tumors, with several biopsies obtained from the tumor section surface for assessing its genetic heterogeneity. Eight of the 12 tumors demonstrated spatial heterogeneity of mutations [2]. A study looking at possible subclonality of a primary tumor demonstrated that as many as eight intratumor areas were required to detect 90% of genomic diversity [3]. At present, personalized anti-BC treatment to a large extent depends on tumor morphology, size, lymph-node metastasis, and the expression of such markers as the estrogen receptor (ER), the progesterone receptor (PR), human epidermal growth factor receptor 2 (HER2), and the cellular marker of proliferation Ki-67. Luminal tumors are ER-positive and represent the most prevalent BC subtype [4]. About 20% of tumors show differences in ER, PR, and HER2 at re-evaluation. The observed differences may have occurred due to both technical factors and intratumoral cellular heterogeneity [5]. Patients with a high level of intratumoral ER heterogeneity had an increased long-term risk of fatal BC [6]. Additionally, it was demonstrated that ER expression was higher at the tumor periphery than in the center [7]. HER2 is a transmembrane tyrosine kinase receptor. There was considerable variation in intratumoral heterogeneity in relation to HER2 copy number between patients because of much shorter recurrence-free survival and because there were fewer survivors in the long term [8]. Ki-67 is a nuclear proliferation marker. It has been demonstrated that it may be of prognostic value in both ER-positive and ER-negative breast carcinomas; however, expression levels of Ki-67 can be higher at the tumor periphery [9]. Apart from genetic heterogeneity, there is epigenetic heterogeneity, which also contributes to tumor intratumoral heterogeneity. MiRNAs are a class of modulators involved in cancer and may be important predictors of disease risk and progression. Quite a few miRNAs showed differential expression between different molecular subtypes of BC [10]. It has been demonstrated that the miRNA profile in breast tumor cells is not the same as in the surrounding morphologically normal tissue. Noteworthy, miRNA expression levels may not be identical in different studies—not even if the tissue type is the same [11]. Results may not be identical due to technical factors, including differences in reagents or types of analysis, sampling protocols, fixation conditions, or storage conditions for unfixed specimens. Dario de Biase et al. compared miRNA expression levels between fresh frozen and paraffin embedded glioblastoma specimens and observed a good correlation [12]. However, Vojtechova et al. showed that the overlap of differentially expressed miRNA between the fresh frozen specimens and the paraffin blocks was only about 30% [13]. Differences in the results of different miRNA expression studies may also be due to intratumoral heterogeneity; however, little is known at the moment about the intratumoral distribution of miRNA expression levels [14].

The aim of this work is to determine the intratumoral distribution of miRNA expression profiles in luminal BC.

## 2. Results

### 2.1. Choosing a Reference Gene for qPCR

For miRNA quantification with RT-qPCR, a reference gene should be chosen. If the expression of the reference gene is variable within the tumor, this will have implications for the 2-∆Ct value. Therefore, the wrong choice of reference gene can be one of the reasons accounting for data inconsistency between different studies. Works exist that analyze miRNA expression levels with normalization to known reference genes (for example, to small nuclear RNA) without testing their expression for stability in the specimens of interest. However, more recent works demonstrated that the expression levels of small nuclear RNA are variable in cancer, and therefore, more and more attention is paid to the fact that not a single gene is universal for all cell types and in all experimental conditions [15]. It is also noted that a reference should have the same properties as has miRNA, because the extraction and identification efficiency for miRNAs may not be the same as that for long noncoding RNA. That is why it is assumed that the best reference gene should be in the same RNA class as that being analyzed. The reference was chosen using geNorm, an algorithm that identifies the most stable genes from among the candidates with measured expression in the specimens of interest [11]. The relative expression levels of miRNA-20a; -21; -125b; -126; -200b; -181a; -205; -221; -222; -451a; -99a; -145; -200a; -214; -30a; -191; and small nuclear RNAs U6, U54, and U58 were measured by RT-qPCR in 132 primary tumor samples and 33 normal mammary gland tissue samples. The optimum number of reference genes was inferred with geNorm (Table 1). This algorithm ranks the genes according to the relative stable expression value denoted by M, where M is the mean pairwise variation of the expression of a gene with that of each of the other control genes. Genes with the lowest M’s have the most stable expression. It is recommended to use the geometric mean of at least three most stable genes to calculate a normalization factor (NF) and the gradual inclusion of more and more control genes until the (n+1)th gene’s contribution to the newly calculated normalization factor NF(n+1) is essential. To find out if more than three genes are required for normalization, we calculated the pairwise variation (Vn/Vn+1) between two successive normalization factors NFn and NF(n+1) for all the genes in question. In our opinion, the best reference in the current settings is the geometric mean of the fluorescence threshold cycles of four miRNAs: miRNA-100, miRNA-143, miRNA-126, and miRNA-125b. Although miRNA-126 and miRNA-125b expression was stable in the sample of the specimens, these miRNAs have some important functions in BC, and hence, we did not include them in the geometric mean of reference genes [16,17]. Further on, the geometric mean of the threshold cycles of miRNA-100 and miRNA-143 will be used as the reference.

### 2.2. Analysis of miRNA Expression Levels between Different Intratumor Areas and Normal Mammary Gland Tissue

The expression levels of 16 miRNAs (miRNA-20a, -21, -125b, -126, -200b, -181a, -205, -221, -222, -451a, -99a, -145, -200a, -214, -30a, -191) were measured by RT-qPCR in four different areas of each tumor (tumor center (C), opposite tumor peripheral sites (P1 and P2), and tumor border(B)) and in normal tissue (N) (Figure 1). Comparative analysis of miRNA expression levels between normal mammary gland tissue and different intratumor areas revealed that 10 miRNAs (miRNA-21, -126, -200b, -221, -222, -99a, -145, -200a, -30a, -191) show differential expression between normal mammary gland tissue and tumor border specimens (*p* < 0.05); 10 miRNAs (miRNA-21, -125b, -200b, -181a, -205, -99a, -145, -200a, -30a, -191) show differential expression between normal mammary gland tissue and central tumor specimens (*p* < 0.05); 9 miRNAs (miRNA -21, -125b, -200b, -181a, -451a, -99a, -200a, -30a, -191); show differential expression between normal mammary gland tissue and tumor periphery 1 (P1) specimens (*p* < 0.05); 13 miRNAs (miRNA-20a, -21, -125b, -126, -200b, -181a, -205, -221, -222, -145, -200a, -214, -191) show differential expression between normal mammary gland tissue and tumor periphery 2 (P2) specimens (*p* < 0.05); Only four miRNAs (miRNA-21, miRNA-200b, miRNA-200a, and miRNA-191) appear as consistently differentiating markers when comparing specimens taken from different intratumor areas and normal tissue (*p* < 0.05) (Table 2). It should be noted that the expression level of any of these four miRNAs is lower in the normal mammary gland tissue than in the tumor specimens. The expression level of miRNA-21 increases with the distance from normal mammary gland tissue; however, neither miRNA-200b nor miRNA-200a nor miRNA-191 was shown to follow this tendency.

### 2.3. Comparative Analysis of miRNA Expression Levels between Specimens Taken from the Tumor Border, Tumor Peripheries, and Tumor Center 

Comparative analysis of miRNA expression levels between tumor border specimens and different intratumor areas revealed that 11 miRNAs (miRNA-20a, -21, -125b, -126, -200b, -181a, -205, -221, -222, -99a, -200a) show differential expression between the tumor border and the tumor center specimens (*p* < 0.05); 8 miRNAs (miRNA-20a, -21, -125b, -126, -221, -222, -451a, -145) show differential expression between the tumor border and the tumor P1 specimens (p<0.05); 13 miRNAs (miRNA-20a, -21, -125b, -181a, -205, -221, -222, -451a, -99a, -145, -200a, -214, -191) show differential expression between the tumor border and the tumor P2 specimens (*p* < 0.05) (Table 3). 

### 2.4. Comparative Analysis of miRNA Expression Levels between Specimens Taken from the Tumor Center and Tumor Peripheries

Comparative analysis of miRNA expression levels between the tumor P1 and the P2 specimens revealed that the expression levels of 13 (miRNA-20a, -21, -125b, -126, -200b, -181a, 205, 221, -222, -451a, -145, -200a, -191) out of 16 miRNAs in question are significantly different (*p* < 0.05) (Table 4). Only miRNA-126 had lower expression levels in tumor P2 than in tumor P1, while those of the other 12 miRNAs were higher in P2 than in P1. Comparative analysis of miRNA expression levels between the tumor center and two tumor peripheral sites revealed that the respective expression levels of miRNA-20a, -21, -125b, -126, -181a, -205, -221, -222, -214, -30a, and -191 are significantly different between the tumor center and the tumor P2 (*p* < 0.05) and that there is no difference in the expression of these miRNAs between the tumor center and the tumor P1. On the other hand, a significant difference has been observed in the expression levels of miRNA-200b, -451a, and -200a between the tumor center and the tumor P1 specimens (*p* < 0.05), while no significant correlation has been found between the respective expression levels of these miRNAs between the tumor center and the tumor P2. Only miRNA-99a and miRNA-145 each showed a statistically significant difference in expression level between the tumor center and either tumor peripheral site (*p* < 0.05).

## 3. Discussion

We looked at the expression levels of 16 miRNAs (miRNA-20a, -21, -125b, -126, -200b, -181a, -205, -221, -222, -451a, -99a, -145, -200a, -214, -30a, -191) in four intratumor areas in each of 33 luminal BC specimens and in surrounding normal mammary gland tissues. Our data suggest that miRNA expression levels are subject to considerable variation, depending on the intratumor area. The data obtained may be interpreted as explaining the inconsistency in miRNA expression estimates in BC coming from different laboratories.

Some works on the role of miRNA-20a in BC show its function as an oncogene with increased expression. Additionally, data published on miRNA-20a expression in different intratumor areas show that it varies across the tumor and that it is lower in tumor than in normal cells [18]. We, too, demonstrated variation in miRNA-20a expression between different intratumor areas; however, the level of expression was higher in the tumor than in the normal specimens and a statistically significant change has been observed only in the BC P2 specimens. A work by Calvano Filho et al. on the expression levels of miRNA-20a in different morphological subtypes of BC reports an average four-fold increase in miRNA-20a expression in triple negative BC as compared with the luminal subtype [19]. However, we observed a four-fold intratumor variation in miRNA-20a expression levels; thus, the choice of a sampling area within the BC tumor matters for the results. 

It has many times been demonstrated that miRNA-125b is an oncosuppressor, and its expression is reduced in the tumor [16]. However, works exist that show miRNA-125b to be an oncogene with enhanced expression and an association with poor prognosis in BC patients and drug resistance [20]. In our study, we observed a reduction in miRNA-125b expression in the tumor P2 and the tumor center as well as its substantial increase in tumor P2 as compared with normal tissue. These data may provide evidence for cellular heterogeneity in tumors as well as for the presence of drug-resistant subclones in them. 

MiRNA-205 is an oncosuppressor and the level of its expression is specific for the BC subtype, the type of the tumor-initiating cell, and the tumor stage. However, there is some controversy: one work demonstrated that miRNA-205 expression is higher in ER+/PR+/HER2+ BC than in any other BC subtypes, while another revealed a high level of miRNA-205 expression in ER−/PR−/HER2- tumors [21]. In our study, we observed a statistically significant intratumoral heterogeneity in miRNA-205 expression in luminal BC. It should be noted that there is a nearly four-fold difference in the expression level of miRNA-205 between the tumor P1 and the tumor P2 areas, which are the most spaced out intratumor locations (*p* < 0.05). 

MiRNA-200a/200b are in the miRNA-200 family and act to suppress tumor development by inhibiting the epithelial-mesenchymal transition [22]. In this work, we observed elevated miRNA-200a and miRNA-200b in tumor tissue, no matter what sampling area, compared with normal tissue. It should be noted that we observed identical patterns of changes in expression for both miRNA-200a and miRNA-200b when comparing different intratumor areas.

Wang et al. found that miRNA-214 was elevated in human BC cell lines and that miRNA-214 overexpression helps avoid apoptosis. It has also been noted that miRNA-214 expression was substantially elevated in BC tissues as compared with the surrounding normal tissues [23]. However, Derfoul et al. showed that miRNA-214 inhibits BC cell proliferation and reduced miRNA-214 expression may facilitate mammary gland tumor development [24]. In our study, we observed increased miRNA-214 expression within the tumor as compared to normal tissues; however, statistically significant differences have been found only for the tumor P2 site.

MiRNA-181a is an oncogene, for it facilitates cell proliferation, its expression is elevated in BC and correlates with poor survival. However, evidence exists for a dual role of miRNA-181a because this miRNA was demonstrated by another work to be an oncosuppressor and a promoter of BC cell apoptosis [25]. MiRNA-21 is a marker of an aggressive BC phenotype. Evidence exists for a correlation between elevated miRNA-21, lymph-node metastasis, and the progressive disease. MiRNA-191 is an oncogene and promotes cell proliferation, migration, and chemoresistance; it is elevated in tumor tissue as compared with the normal mammary gland tissue [26]. Elevated expression of the miRNA-221/222 cluster promotes cancer cell proliferation to form an invasive phenotype and is a predictor of poor prognosis in BC patients [27]. In this work, the expression levels of the miRNA-181a, miRNA-21, and miRNA-191 oncogenes, and oncogenes-221/222 are higher within any intratumor specimen than in normal tissue; however, there are statistically significant differences in their expression levels between intratumor areas.

MiRNA-126 is antiangiogenic and its expression is reduced in BC cells [17]. MiRNA-99a is an oncosuppressor and inhibits cell proliferation and invasion, promoting apoptosis [28]. MiRNA-145 is an oncosuppressor, and its expression is reduced in BC cells [29]. We, too, observed a tendency towards a reduction in the expression levels of these miRNAs in tumor specimens as compared to normal tissue; however, we found statistically significant differences in expression not only between the tumor and the normal tissue, but also between intratumor areas. Noteworthy, we observed elevated miRNA-145 expression in the tumor P2 as compared with normal tissue. MiRNA-30a is another oncosuppressor. It inhibits invasive BC, and its reduced expression is associated with poor prognosis in BC patients [30]. We observed elevated miRNA-30a in tumor tissue, no matter what sampling area, compared with normal tissue. Our results may be explained by the fact that we included in the study the luminal BC subtype, which is noted for the best prognosis.

Some miRNAs, which suppress the expression of antioncogenes play roles as oncomirs, and their expression is increased in breast cancer [31]. Other miRNAs are associated with directing their suppressor potential in breast cancer [32]. High expression of miRNA-20a significantly decreased the mRNA and protein levels of RUNX3, as well as its direct downstream targets Bim and p21. Overexpression of miRNA-20a promoted the migration and invasion of breast cancer cells in vitro [33]. MiRNA-221, miRNA-21 and miRNA-222 displayed oncogenic roles through negative regulation of PTEN [32,34]. Overexpression of miRNA-99a and decreased expression of CDC25A could suppress breast cancer cell proliferation and invasion [35]. MiRNA-30a inhibits breast cancer proliferation and metastasis by directly targeting MTDH [36]. The miR-200 family suppresses EMT by the regulation of E-cadherin [37]. Downregulation of miRNA-451a upregulated MIF expression and increased breast cancer cell growth, invasion, and tamoxifen sensitivity [38]. Other results establish the regulation of MMP-14 in cancers by miRNA-181a, and they further suggest strategies to elevate miRNA-181a to prevent cancer metastasis [39]. 

We demonstrated that all the miRNAs in question are differentially expressed between normal mammary gland tissue and different intratumor areas; however, only miRNA-125b, miRNA-181a, miRNA-21, miRNA-200b, miRNA-200a, and miRNA-191 are seen to be consistently differentiating markers when comparing specimens taken from all intratumor areas in question and normal tissue (*p* < 0.05) (Table 2). Thus, no matter what the sampling area is, it will not affect the measured expression of these miRNAs. Noteworthy, a comparative analysis of miRNA expression levels between normal mammary gland tissue and their border revealed statistically significant differences for 10 miRNAs, suggesting possible distinctions in the morphological and genetic features of the tumor border.

After comparing the tumor P1 and tumor center specimens, we found statistically significant differences in expression between five miRNAs, and after comparing the tumor P2 and tumor center specimens, differences were observed for 12 miRNAs. At this point, we cannot say what exactly makes the tumor P2 specimens so different from the tumor P1 ones that the differences in miRNA expression are that great. Again, P1 and P2 are the most spaced out intratumor locations. It is possible that this special configuration of the specimens can explain the highest heterogeneity in expression levels among all miRNAs being discussed.

## 4. Materials and Methods 

### 4.1. Materials

The study included 33 certain BC cases of the luminal A or luminal 2 (Her2-) subtypes. The patients had received no adjuvant therapy before surgery. The study material was in the form of tumor and adjacent morphologically unchanged tissue imprinted on the slides. For each case, tumor imprints were taken from four different tumor areas (tumor center, opposite tumor peripheral sites, and tumor border) and a normal tissue imprint from adjacent morphologically unchanged tissue. Permission for this study was granted by the Biomedical Research Ethics Committee in the Kulakov Federal Research Center for Obstetrics, Gynecology, and Perinatology of the Ministry of Health of the Russian Federation. The research was carried out under the law of the Russian Federation, the ethical norms and principles set out in the Declaration of Helsinki (1964), with all additions and amendments in relation to scientific research using human biomaterial, and according to the International Ethical Guidelines for Biomedical Research Involving Human Subjects published by the Council for International Organizations of Medical Sciences (CIOMS). All the patients’ source data were anonymized in accordance with the requirements set out in clause 3 of Article 6 of the current Russian Federal Law on Personal Data (No. 152-FZ). Written informed consent was obtained from each patient. Biomaterial was sampled according to the Standard Operating Procedure (SOP) protocol developed by the Kulakov Federal Research Center for Obstetrics, Gynecology, and Perinatology of the Ministry of Health of the Russian Federation in accord with Order of the Ministry of Health of the Russian Federation No. 179н of March 24, 2016 “On the Rules for Pathoanatomical Research” and the clinical recommendations laid down in the “Standard Technological Procedures in Pathoanatomical Research” by the Russian Society of Pathologists.

### 4.2. RNA Extraction 

Nucleic acids were extracted from specimens using the RealBest Extraction 100 Reagent Kit (Vector-Best, Novosibirsk, Russia). Mammary gland tissue was washed off the slide with 600 µL of guanidine lysis buffer. The suspension was mixed vigorously in a TS-20 thermo-shaker (Biosan, Riga, Latvia) for 15 min at 65 °C. The solution was then centrifuged for 2 min at 10,000 rpm (Eppendorf MiniSpin, Humburg, Germany), the supernatant was transferred to fresh tubes and supplemented with an equal volume of isopropanol and 10 µL of magnetic bead suspension. That was mixed and allowed to stay for 4 min at room temperature. At the next extraction stage, the solution was centrifuged for 10 min at 13,000 rpm (Eppendorf MiniSpin, Humburg, Germany), the supernatant was removed, the pellet was washed with 500 µL of 70% ethanol and then with 300 µL of acetone. The precipitate was dried and dissolved in 200 µL of eluting solution.

### 4.3. Reverse Transcription

The reverse transcription reaction for cDNA was carried out in a volume of 30 µL. Ready-for-use reactions RealBest RT Master Mix (Vector-Best, Novosibirsk, Russia) were utilized. The reverse transcription reaction contained 3 µL of extracted RNA, 16.2 µL of 40% threhalose solution, 3 µL of 10× reverse transcription buffer, 3 µL of 4 mM deoxynucleotide triphosphate solution, 3 µL of 10% bovine serum albumin solution (BSA) solution, 0.32 µL of reverse transcriptase (Vector-Best, Novosibirsk, Russia), and 1.5 µL of 10 µM primer solution for reverse transcription. All oligonucleotides were synthesized by Vector-Best (Novosibirsk, Russia). Oligonucleotides were selected using the PrimerQuest online service (http://eu.idtdna.com/). The sequences of primers and fluorescently labeled probes are available in Appendix A. Three microliters of the reaction mixture containing cDNA was used immediately as a template for a real-time PCR on a CFX96 system (Bio-Rad, California, USA). 

### 4.4. Real-time PCR

MiRNA expression levels were measured by real-time PCR on a CFX96 amplifier (Bio-Rad Laboratories, California, USA). The reaction was carried out in a volume of 30 µL containing 3 µL of cDNA, 14 µL of H2O, 3 µL of 10× PCR buffer (Vector-Best), 3 µL of 4 mM deoxynucleotide triphosphate solution, 3 µL 10% BSA solution, 1 µL of Taq polymerase (Vector-Best, Novosibirsk, Russia) along with monoclonal antibodies to its active center (Clontech, California, USA), 3 µL of a mix of forward and reverse primer (5 µM) and probe (2.5 µM). The primers and the probes are Vector-Best (Novosibirsk, Russia) developments, the efficiency of the PCR being 85%–100%. Analysis of the threshold cycles generated by the qPCR was performed using the 2-∆Ct method. Statistical processing of data was carried out using STATISTICA v12.0 (StatSoft Inc., OK, USA). Two independent samples were compared using the Mann–Whitney U test.

### 4.5. GeNorm Algorithm

The geometric mean of the quantification cycles of two miRNAs (NF2) was calculated by Formula (1):(1)$$C_{q}\left(NF2\right)=\sqrt{C_{q1}*C_{q2}}$$

The most stable reference genes were selected using geNorm algorithm: let there be data on the expression n of various miRNAs in m samples; for each pair of miRNAs, the vector *A_jk_* was calculated, the components of which are computed as the logarithm to the base two of the ratio between the level of miRNA expression in a single sample (Formula (2)):(2)$$A_{jk}=\left\{log_{2}\left(\frac{a_{1j}}{a_{1k}}\right),log_{2}\left(\frac{a_{2j}}{a_{2k}}\right)\;,\;\ldots ,\;log_{2}\left(\frac{a_{mj}}{a_{mk}}\right)\;\right\}\;,\;\forall_{j,k}\;\in \left[1,n\right],\;k\neq j$$

The pairwise variation *V_jk_* of miRNA defined as standard derivation of *A_jk_* elements (Formula (3)):(3)$$V_{jk}=\sum_{k=1}^{n}st.dev\left(A_{jk}\right)\;,\;k\neq j$$

The expression stability of miRNA _j_ (*M_j_*) is the arithmetic mean of all pairwise variations *V_jk_* (Formula (4)).
(4)$$M_{j}=\frac{\sum_{k=1}^{n}V_{jk}}{n-1}\;,\;k\neq j$$

## Figures and Tables

**Figure 1 ncrna-06-00016-f001:**
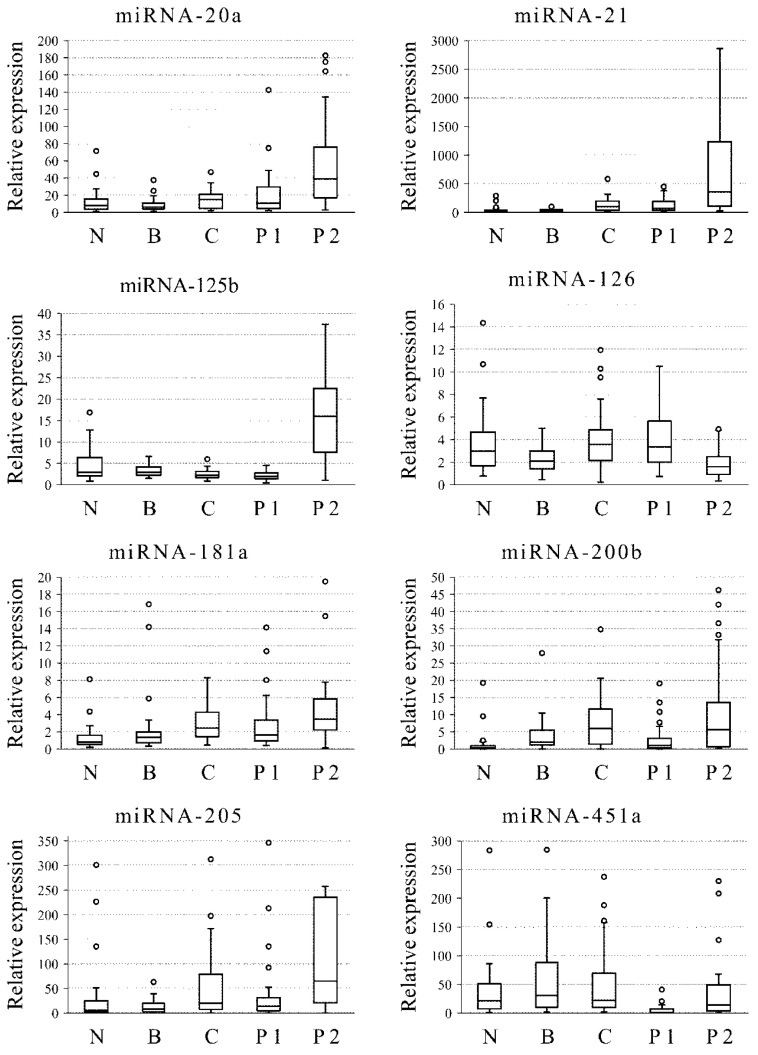
Comparative analysis of miRNA expression levels between specimens taken from different intratumor areas: tumor center (C), two tumor peripheries (P1 and P2), and tumor borders (B) with normal mammary gland tissue (N). Shown are the median value, the upper and the lower quartiles, the outlier-free range, and outliers (appear as circles).

**Table 1 ncrna-06-00016-t001:** Assessing gene stability with geNorm to determine the optimum number of reference genes.

miRNA	Stability Measure (M)	V Name	NFn/NF(n+1) Variation	NFn Stability
miRNA-143	1.707295837		1	
miRNA-100	1.80022993	V1/2	0.528235232	1.670849973
miRNA-126	1.800986451	V2/3	0.361285079	1.63594884
miRNA-125b	1.811601327	V3/4	0.2027059	1.641631035
miRNA-145	1.859214933	V4/5	0.16494292	1.650351674
miRNA-20a	1.876414282	V5/6	0.220042327	1.618486051
miRNA-21	1.933340528	V6/7	0.203421143	1.591868665
miRNA-222	1.937542742	V7/8	0.147144095	1.58906537
miRNA-181a	1.969174988	V8/9	0.139065169	1.584626526
miRNA-221	1.971881076	V9/10	0.113726124	1.58903484
miRNA-204	2.00646399	V10/11	0.119749575	1.583226837
miRNA-214	2.106526407	V11/12	0.109896221	1.587798283
miRNA-30a	2.156076925	V12/13	0.136245315	1.570279014
U6	2.159702998	V13/14	0.123657069	1.558184739
U58	2.256085709	V14/15	0.118561989	1.553964734
miRNA-191	2.307850884	V15/16	0.117821227	1.548875535
miRNA-99a	2.336822525	V16/17	0.09952473	1.551560708
miRNA-200a	2.380047394	V17/18	0.112528959	1.545511535
U54	2.391901029	V18/19	0.098671803	1.548786269
miRNA-200b	2.70503197	V19/20	0.116023318	1.552206252
miRNA-205	2.808484382	V20/21	0.112807048	1.552130251
miRNA-451a	3.301276521	V21/22	0.13524655	1.551292987

**Table 2 ncrna-06-00016-t002:** Significance level of differences in miRNA expression levels between normal (*n* = 33) and different intratumor areas (*n* = 132).

miRNA	Normal vs. Tumor Border	Normal vs. Tumor Center	Normal vs. Tumor Periphery1	Normal vs. Tumor Periphery2
miRNA-20a	0.615703	0.062666	0.130845	0.000003 *
miRNA-21	0.019136 *	0.000001 *	0.000006 *	0.000000 *
miRNA-125b	0.561720	0.018225 *	0.001328 *	0.000002 *
miRNA-126	0.017146 *	0.683480	0.750119	0.001682 *
miRNA-181a	0.057594	0.000003 *	0.001461 *	0.000000 *
miRNA-200b	0.000029 *	0.000000 *	0.030546 *	0.000004 *
miRNA-205	0.911808	0.031572 *	0.112178	0.000014 *
miRNA-221	0.013697 *	0.779285	0.888576	0.000000 *
miRNA-222	0.011748 *	0.702299	0.898649	0.001328 *
miRNA-451a	0.281086	0.18650	0.000000 *	0.344958
miRNA-99a	0.0000001 *	0.000000 *	0.000418 *	0.081107
miRNA-145	0.002129 *	0.020995 *	0.637295	0.000008 *
miRNA-200a	0.000027 *	0.000000 *	0.000011 *	0.000000 *
miRNA-214	0.094485	0.959369	0.407011	0.001531 *
miRNA-30a	0.018452 *	0.001095 *	0.001934 *	0.183532
miRNA-191	0.000425 *	0.012171 *	0.002541 *	0.000000 *

* statistically significant differences are in bold (*p* < 0.05).

**Table 3 ncrna-06-00016-t003:** Significance level of differences in miRNA expression between specimens taken from intratumor areas and border.

miRNA	Tumor Border vs. Tumor Center	Tumor Border vs. Tumor Periphery1	Tumor Border vs. Tumor Periphery2
miRNA-20a	0.030221 *	0.038173 *	0.000001 *
miRNA-21	0.000029 *	0.000155 *	0.000000 *
miRNA-125b	0.004383 *	0.000195 *	0.000000 *
miRNA-126	0.005207 *	0.008225 *	0.226516
miRNA-181a	0.001454 *	0.197599	0.000067 *
miRNA-200b	0.012689 *	0.068964	0.099840
miRNA-205	0.015337 *	0.123681	0.000000 *
miRNA-221	0.002448 *	0.020571 *	0.000000 *
miRNA-222	0.012689 *	0.004383 *	0.000001 *
miRNA-451	0.378257	0.000000 *	0.040743 *
miRNA-99a	0.000207 *	0.911808	0.008915 *
miRNA-145	0.193051	0.005912 *	0.000290 *
miRNA-200a	0.000001 *	0.066952	0.000000 *
miRNA-214	0.252791	0.799400	0.022097 *
miRNA-30a	0.114274	0.111264	0.350545
miRNA-191	0.414709	0.984406	0.000001 *

* statistically significant differences are in bold (*p* < 0.05).

**Table 4 ncrna-06-00016-t004:** Significance level of differences in miRNA expression between specimens taken from intratumor areas.

miRNA	Tumor Center vs. Tumor Periphery1	Tumor Center vs. Tumor Periphery2	Tumor Periphery1 vs. Tumor Periphery2
miRNA-20a	0.918841	0.000077 *	0.000946 *
miRNA-21	0.798904	0.000054 *	0.000057 *
miRNA-125b	0.179320	0.000000 *	0.000000 *
miRNA-126	0.908738	0.000572 *	0.000815 *
miRNA-181a	0.127579	0.137569	0.006775 *
miRNA-200b	0.000156 *	0.721300	0.003172 *
miRNA-205	0.338451	0.004479 *	0.000147 *
miRNA-221	0.888576	0.000000 *	0.000000 *
miRNA-222	0.601311	0.001266 *	0.006247 *
miRNA-451a	0.000000 *	0.239813	0.000011 *
miRNA-99a	0.000357 *	0.000004 *	0.068390
miRNA-145	0.035976 *	0.000042 *	0.000002 *
miRNA-200a	0.007054 *	0.062666	0.000069 *
miRNA-214	0.358206	0.006506 *	0.053998
miRNA-30a	0.858470	0.042196 *	0.106438
miRNA-191	0.319401	0.000000 *	0.000004 *

* statistically significant differences are in bold (*p* < 0.05).

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
