# Peer review of "Intratumoral Heterogeneity of Expression of 16 miRNA in Luminal Cancer of the Mammary Gland"

_ncrna, 2020, doi:10.3390/ncrna6020016_

Round 1

Reviewer 1 Report

The manuscript is well written and the study addresses an interesting and current problem about the role of miRNA on Breast cancer.
However, I have certain minor suggestions related to the paper.

Sometimes you use microRNA and sometime miRNA, try to use the same word in the text.

In figure 1, please explicit the meaning of the term “N”, “B”, “C”, “P1”, “P2” both in the legend and in the main text (whereas only some of these are present).

Line 121 you wrote”-191;” instead of “-191)”

Table 2, table3, table 4: at the end you write: “*statistically significant differences are in bold (p<0.05)”. Nothing is in “bold” except the heading of each column! In this way, the results in the tables are difficult to follow.

The “Materials and Methods” section and “Discussion” must be formatted as “justified”.

Please resubmit after the issues have been fixed.

Author Response

1) MicroRNA-miRNA: I fixed the error in 22 places.

2) Comparative analysis of miRNA expression levels between specimens taken from different intratumor areas: tumor center (C), two tumor peripheries (P1 and P2) and tumor borders (B) with normal mammary gland tissue (N).

3) The expression levels of 16 miRNAs (miRNA-20a; -21; -125b; -126; -200b; -181a; -205; -221; -222; -451a; -99a; -145; -200a; -214; -30a; -191) were measured by RT-qPCR in four different areas of each tumor (tumor center (C), opposite tumor peripheral sites (P1 and P2), and tumor border(B)) and in normal tissue (N) (Fig.1).

4) ....-200b; -181a; -205; -221; -222; -145; -200a; -214; -191) 

5) Statistically significant differences are in bold (p<0.05)

6) The “Materials and Methods” section and “Discussion” are formatted as “justified”.

Reviewer 2 Report

This is an observational study in which the authors analyzed the expression of 16 select miRNAs, along with U6, U58, and U54 in breast tumor samples.  They report that miR-143 is the best normalizer since it was the most stable in expression. They then examined the expression of 16 miRNAs in 4 areas of the breast tumors (center, periphera, border) vs. normal breast and identified site differences. 

Major comment: They did not examine the expression of any targets of the location-specific miRNAs, e.g., increased miR-21 in P2, does that correlate with downregulation of targets, e.g., PTEN?

Specific comments:

  1. Table 1 legend: Please add the number of normal and breast tumors analyzed. Add how M, V, and NFn are calculated.  Readers want to obtain this information without reading the text.
  2. Tables1, 2, 3: Two decimal points should be used. Also, please use 2.00 vs. 2,00 (no comma, use a period). 
  3. Tables 1,2, 3: While the authors are using bold to denote significant differences, that can be difficult to see, so please add * to denote significance.
  4. Did the central areas of the tumors show necrosis?
  5. Is there a difference in miRNA location in the tumor with ER/PR, HER2, or TNBC status?
  6. With respect to miR-143 as normalizer – miR-143 was reported to be downregulated in breast cancer, involved in RAS-signaling, and has targets in breast cancer progression. This has been reviewed and should be commented on see https://www.mdpi.com/2311-553X/4/4/40
  7. Likewise in the Discussion, the authors should cite reviews on the roles of the 16 miRNAs and their targets in breast cancer.
